# Research Progress on the Structural and Functional Roles of hnRNPs in Muscle Development

**DOI:** 10.3390/biom13101434

**Published:** 2023-09-22

**Authors:** Zhenyang Li, Haimei Wei, Debao Hu, Xin Li, Yiwen Guo, Xiangbin Ding, Hong Guo, Linlin Zhang

**Affiliations:** Key Laboratory of Animal Breeding and Healthy Livestock Farming, College of Animal Science and Veterinary Medicine, Tianjin Agricultural University, Tianjin 300392, China; 2103010106@stu.tjau.edu.cn (Z.L.); 2203010117@stu.tjau.edu.cn (H.W.); hudebao@tjau.edu.cn (D.H.); lixin82@tjau.edu.cn (X.L.); yiwenguo@tjau.edu.cn (Y.G.); xiangbinding@tjau.edu.cn (X.D.); guohong@tjau.edu.cn (H.G.)

**Keywords:** hnRNPs, alternative splicing, muscle development, muscle disorders

## Abstract

Heterogeneous nuclear ribonucleoproteins (hnRNPs) are a superfamily of RNA-binding proteins consisting of more than 20 members. These proteins play a crucial role in various biological processes by regulating RNA splicing, transcription, and translation through their binding to RNA. In the context of muscle development and regeneration, hnRNPs are involved in a wide range of regulatory mechanisms, including alternative splicing, transcription regulation, miRNA regulation, and mRNA stability regulation. Recent studies have also suggested a potential association between hnRNPs and muscle-related diseases. In this report, we provide an overview of our current understanding of how hnRNPs regulate RNA metabolism and emphasize the significance of the key members of the hnRNP family in muscle development. Furthermore, we explore the relationship between the hnRNP family and muscle-related diseases.

## 1. Introduction

Various post-transcriptional modifications are required for the maturation of mRNA in eukaryotic cells. These modifications include the addition of 7-methylguanosine (m7G) at the 5′ end, the formation of a polyadenylic acid tail at the 3′ end, and RNA splicing. Alternative splicing allows for the production of different mature mRNAs from a single pre-mRNA molecule. Heterogeneous nuclear ribonucleoproteins (hnRNPs) are a superfamily of RNA-binding proteins that play a key role in regulating the alternative splicing of pre-mRNA [1]. In recent years, the role of hnRNPs in muscle biology has gained attention. Several hnRNPs have been implicated in myogenesis, and their association with various muscle diseases has also been discovered. This review provides an overview of the structure and function of the hnRNPs protein family, focusing on their role in myogenesis and muscle disease processes.

## 2. Overview of hnRNPs

### 2.1. Composition of hnRNPs

Heterogeneous nuclear ribonucleoproteins (hnRNPs) are a class of RNA-binding proteins (RBPs). Through immunopurification and two-dimensional gel electrophoresis, 42 kinds of hnRNPs have been identified in Hela cells. These hnRNPs consist of more than 20 major hnRNPs that exhibit relatively higher abundance, along with other hnRNPs [2,3]. The main hnRNPs have a molecular weight ranging from 34 kDa to 120 kDa and are named hnRNP A1 to hnRNP U [4,5,6,7] based on their molecular weight and structural and functional characteristics. Due to the strong association between hnRNP A1, A2/B1, B2, C1, and C2 and hnRNA, they are referred to as “core proteins” [8].

### 2.2. hnRNPs: RNA-Binding Domains and Structural Insights

The analysis of cDNA sequences has revealed that hnRNPs possess a modular structure comprising at least one RNA-binding motif and auxiliary domains [9,10] (Figure 1). As recent investigations into the structure of hnRNPs have delved into greater detail, it has been confirmed that hnRNPs consist of three distinct RNA-binding motifs: the RNA recognition motif (RRM) [11,12], the RNA-binding domain consisting of Arg-Gly-Gly repeats (RGG domain) [13], and the K-homology domain (KH domain) [14].

Researchers have conducted comprehensive analyses of the sequences of RNA-binding motifs and the auxiliary domains of hnRNPs, resulting in a well-established understanding of the primary functions associated with each motif. Among these, the RNA recognition motif (RRM) stands out as the most prevalent RNA-binding domain within hnRNPs. Comprising 80–90 amino acid residues, the RRM is characterized by highly conserved RNP1 ([K/R]-G-[F/Y]-[G/A][F/Y]-[I/L/V]-X-[F/Y]) and RNP2 ([I/L/V]-[F/Y]-[I/L/V]-X-N-L) amino acid sequences [14,15]. A typical RRM is composed of a four-stranded anti-parallel β-sheet that folds into two α-helices, forming the characteristic β1α1β2β3α2β4 topology [16]. Functionally, the RRM acts as an RNA binding platform, playing a pivotal role in the specific binding of hnRNPs to pre-mRNA [12]. Studies have revealed that the RNA binding of the RRM often relies on the aromatic amino acid residues within RNP1 (β1) and RNP2 (β3), as well as the first basic amino acid residue of RNP1 [17]. For instance, in hnRNP A1, the RRM domain binds to RNA (UAGGG(A/U)) via aromatic amino acid residues located on the β-sheet [18]. Interestingly, the RRM domain of hnRNP F employs three highly conserved loop structures (loop 1, 3, and 5) to interact with AGGGAU, deviating from the traditional RNP1 and RNP2 binding mode [19]. The β-sheet of hnRNP F’s RRM domain does not engage with RNA [20]. Currently, a diverse array of RRM-RNA complexes have been identified within cells with the ability to predict the RNA segments they recognize [21]. Many hnRNPs also harbor multiple RRM domains, where their role in RNA binding within hnRNPs extends beyond mere summation, intricately entwined with the elaborate conformations they adopt [18]. Only a handful of hnRNPs lack RRM domains, including hnRNP E1-E4, hnRNP J/K, and hnRNP U.

RGG also has RNA binding activity [22]. It is characterized by closely spaced arginine–glycine–glycine tripeptide repeat clusters and interspersed aromatic amino acid residues [23]. The RGG box contains many arginines which carry positive charges. These residues can form electrostatic interactions with non-specific RNA to enhance RNA binding. Additionally, they can form a hydrogen bond network with specific RNA to specifically recognize the tertiary structure of RNA, such as stem-loop or convex structures [22].

The KH domain consists of 70 amino acids. Within the KH domain, the conserved GxxG loop plays a crucial role in recognizing and binding RNA bases through hydrogen bonds and hydrophobic interactions [24,25,26]. These interactions allow the KH domain to specifically recognize and bind certain RNA sequences. Additionally, KH domains have the ability to recognize longer RNA targets in a tandem manner [14].

Furthermore, certain studies [22,27,28] have demonstrated the ability of these RNA-binding motifs to bind to DNA. This suggests that certain hnRNPs may engage in DNA binding to execute additional functions, although this aspect will not be further explored in this study.

It is worth noting that hnRNP proteins also contain other domains that mediate important functional specificities. The most common and well-characterized auxiliary domains are the nuclear localization signal (NLS) and M9 domains, both of which are associated with nucleoplasmic shuttling [29,30]. hnRNP A2/B1, C, Q, and R all contain NLS motifs, and NLS binds to the nuclear transfer factor importin α-β heterodimer to facilitate nucleoplasmic shuttling through the nuclear pore complex. Both hnRNP A1 and hnRNP A2/B1 contain the M9 domain, which enables nuclear localization by binding to the transport receptor Trn1 [31,32]. Additionally, hnRNP U also possesses the SAP domain related to DNA binding (Scaffold Attachment Factor A/B Acinus PIAS domain) [33,34] and the ATPase domain to regulate the oligomerization of chromatin-associated RNAs (caRNAs) [35,36]. However, the precise mechanisms of action for these two domains are still unclear. The function of hnRNPs is dependent on their domains and binding affinity to pre-mRNA. In their study, Massimo Caputi et al. [37] discovered that the core binding site of the RRM domain of the hnRNP H family (H, H’, F, 2H9) is GGGA. Additionally, high-affinity binding sequences for hnRNP A1 [38] and C [39] have been identified through selection and amplification from random RNA libraries.

### 2.3. Functions of hnRNPs

Due to the diverse types and complex structural domains of hnRNPs, they can perform a variety of functions in cells. These functions include processing pre-mRNA into mature mRNA and serving as *trans-acting* factors that regulate gene expression. hnRNPs also participate in the processing of pre-mRNA alongside other RNPs, playing a crucial role in regulating mRNA transport, localization, translation, and stability (Figure 2). As a result, hnRNPs play a key role in various biological processes within cells. Contemporary research on hnRNPs primarily focuses on their involvement in muscle and neurodegenerative diseases (such as amyotrophic lateral sclerosis) as well as their role in cancer progression [40,41,42].

#### 2.3.1. Regulation of Alternative Splicing

The pre-mRNA splicing process in cells relies on the interaction between splicing factors and splice sites. This mainly includes the following steps: (1) The U1 snRNP complex binds to the 5′ splice site [43,44]. (2) U2AF65 binds to the polypyrimidine site (Py-tract) [45]. (3) U2AF35 binds to the AG bases at the 3′ splice site. (4) The U2AF heterodimer binds to the 3′ splice site [46] (Figure 3). Most hnRNPs typically function as inhibitory splicing regulators, and their mechanism is as follows.

Splice site identification and shelter: hnRNP proteins have the ability to bind to pre-mRNA splice sites, which can impact the formation of spliceosomes. This binding can mask or hinder the recognition of splice sites, leading to differential alternative splicing. Additionally, the binding of hnRNPs can competitively affect the binding of other splicing factors. For instance, hnRNP I’s RRM1 and RRM2 specifically bind to the polypyrimidine sequence of the fourth internal loop of the U1 snRNA stem loop, thereby inactivating the splicing complex A formed by U1 snRNP [47]. Studies have also shown that hnRNP I can target *ITSN1* [48], *β-tropomyosin* [49], *Fas* [50], and *alpha-actinin* [51], competitively inhibiting the binding of U2AF65 and preventing the formation of the U2AF heterodimer complex by binding to the polypyrimidine sequence of pre-mRNA. hnRNP A1 interacts with the 3′ splice site of *MAPT* exon 10 and facilitates the exclusion of exon 10 [52]. Similarly, hnRNP L can bind to pre-mRNA polypyrimidine sequences, competitively inhibiting the binding of U2AF65 to RNA [53]. Furthermore, hnRNP H1/H2 counteracts the activation of the 3′ splice site [54] (Figure 4B).Splicing inhibition: hnRNPs can impact RNA structure by interacting with pre-mRNA, resulting in the exclusion of specific exons. This inhibitory effect primarily operates by altering the structure of the splice site. For instance, the RRM3 and RRM4 domains of hnRNP I bind to the polypyrimidine sequence (e.g., CUCUCU) near the pre-mRNA exon, forming an RNA ring structure that hinders the binding of other splicing factors and the formation of splicing complexes [51,55] (Figure 4C).Competitive splicing inhibition: In certain cases, hnRNP proteins and other RNA-binding proteins may competitively bind to the same pre-mRNA, which can impact the inclusion or exclusion of specific exons. The majority of competitive splicing occurs between hnRNPs and SRs. For instance, hnRNP H1 can compete with SRSF3 for binding to *PRMT5* pre-mRNA, thereby inhibiting the exclusion of *PRMT5* exon 3 by SRSF3 [56]. Additionally, hnRNPA1 can competitively bind to the G-rich sequence downstream of *β-tropomyosin* exon 6B, along with ASF/SF2, leading to the inhibition of exon 6B exclusion. Simultaneously, hnRNP A1 and ASF/SF2 competitively bind to the 5′ splice site of C175G pre-mRNA (C175G is a synthetic 533 nt pre-mRNA sequence that is frequently employed as a standard model for investigating 5′ splice sites). hnRNP A1 competitively inhibits the binding of U1 snRNP to the 5′ splice site of C175G pre-mRNA, while ASF/SF2 enhances the binding of U1 snRNP to the 5′ splice site of C175G pre-mRNA [57] (Figure 4D).

Recent studies have demonstrated that hnRNPs can also function as splicing enhancers, thereby promoting alternative splicing. For instance, SCHAUB M et al. [58] discovered that members of the hnRNP H family play a crucial role in the alternative splicing of *HIV-1 tat* pre-mRNA by acting as splicing enhancers. Additionally, hnRNP K has been found to bind downstream of the 5′ splice site of *IAV* pre-mRNA, facilitating the recruitment of U1 snRNP [59]. The underlying reason for these observations may be attributed to the binding site of hnRNPs [60]. The mislocalization of hnRNPs on the opposite side of the 5′ splice site leads to an increased recruitment of U1 snRNP and promotes splice site recognition [61,62].

#### 2.3.2. Regulation of mRNA Stability

hnRNPs play a role in regulating mRNA stability through various mechanisms, including the poly(A) tail, AU-rich elements (AREs), and the 3′UTR. For instance, hnRNP H1 and hnRNP F can enhance the stability of *APP* mRNA by binding to it through cytoplasmic shuttling mediated by the gly Rich motif, thereby increasing its half-life [63]. Additionally, hnRNP F has been found to regulate the TTP/BRF-mediated degradation of ARE-mRNAs [64]. The RRM2 domain of hnRNP A2/B1 acts as an m6A reader [65], recognizing the N6-methyladenosine (m6A) site on *TCF7L2* mRNA to stabilize the poly(A) tail and maintain its mRNA stability. Moreover, hnRNP A2/B1 and hnRNP A1 play a role in the regulation of mRNA deadenylation via the CCR4-NOT deadenylation complex. They achieve this by binding to the UAASUUAU sequence present in the mRNA 3′UTR, thereby influencing the degradation of the transcript [66] (Figure 5B). Interestingly, hnRNP A2/B1 also binds to its own 3′UTR to regulate the ratio of nonsense-mediated RNA decay (NMD) (both for sensitive and insensitive types). When the levels of hnRNP A2/B1 protein increase, it combines with its own pre-mRNA 3′UTR, leading to alternative splicing and the production of a higher proportion of NMD-sensitive mRNAs. These NMD-sensitive mRNAs are subsequently degraded during translation [67]. This auto-regulatory mechanism demonstrates how splicing factors control their own expression levels, providing one possible explanation [68]. As an ARE-binding protein, hnRNP D has the ability to recognize and bind to various ARE-mRNA sequences by targeting uridine residues [69]. Subsequently, it recruits transporters such as eIF4G, PABP, Hsp70, Hsc70, and Hsp27 to form the signal transduction regulatory complex (ASTRC). ASTRC plays a crucial role in initiating the 3′ to 5′ deadenylation-dependent mRNA degradation pathway, thereby promoting mRNA degradation. Among the isoforms of hnRNP D, the specific mechanism of action involves AUF1 interacting with the AU-rich element (ARE) in mRNA, which serves as a recognition site for AUF1. While the mRNA is being translated, AUF1 interacts with the translation initiation factor eIF4G [70], causing AUF1 to be released from the ARE by the ribosome. This enables the ribosome to access the mRNA and initiate translation. Simultaneously, AUF1 forms a complex with the polyadenine nucleic acid-binding protein (PABP), which exposes the polyadenine nucleic acid tail and allows the mRNA to be degraded by nuclease. The p40 isoform is involved in the decay of ARE-mRNA, which regulates lymphokine mRNA stability [71] (Figure 5A). Additionally, studies have demonstrated [72] that the KH3 domain of hnRNP K can bind to the poly(C) site of LAPTM5 3′UTR, thereby enhancing the stability of its transcripts. Different hnRNPs exhibit variations in their regulation of mRNA stability. For instance, hnRNP D, K, I, and Q [73] can all bind to the 3′UTR of mPer3, but they differ in their impact on mPer3 stability. hnRNP K maintains the stability of mPer3, while hnRNP D and Q accelerate its degradation. hnRNP I, on the other hand, does not affect the stability of mPer3. Additionally, hnRNP A1 [74], C [75], U [76], and L [77] can bind to the 3′UTR of certain mRNAs and influence their stability, although the mechanism underlying this remains unclear.

#### 2.3.3. Localization and Transport of mRNAs

RNA molecules transported within cells often contain specific cis-acting elements that are recognized by specific trans-acting factors in the cell. The interaction between these cis-acting elements and trans-acting factors, along with the involvement of molecular motors, enables RNA particles to actively transport on microtubules and actin filaments. The hnRNP AB and hnRNP A2 proteins contain RRM motifs that can bind to the cis-acting element RTS on the 3′UTR of mRNA [78]. This binding facilitates the transfer of mRNA from the nucleus to the cytoplasm. During this process, hnRNP AB may target specific transcripts by stabilizing RNA G4 quadruplexes near the transcript’s RTS. Several transcripts, including MBP, β-actin, Arc, BDNF, CAMKIIα, and Protamine 2, require hnRNP AB for their localization and transport [79,80,81,82,83]. The glycine-rich domain (GYR) of the hnRNP H/F protein interacts with Transportin 1 to facilitate nucleocytoplasmic shuttling and participate in the extranuclear transport of mRNA [84]. The M9 domain of hnRNP A1 is considered the main functional domain for nucleocytoplasmic shuttling, and its nuclear import does not rely on the classic NLS pathway [85].

## 3. Role and Function of hnRNPs in Regulating Muscle Development

Muscle development is a highly intricate process that involves several stages: muscle satellite cell proliferation and fusion to form myotubes, the differentiation of myotubes into muscle fibers, and, ultimately, the maturation and development of these fibers into muscles. These processes are influenced by numerous regulatory factors. Recent research has identified the significance of hnRNP A/B, hnRNP M, hnRNP E1, hnRNP G, hnRNP L, and hnRNP H in muscle development.

### 3.1. Role of hnRNP A/B during Muscle Development

hnRNP A/B is a core member of the hnRNP family that consists of four subtypes: hnRNP A0, hnRNP A1, hnRNP A2/B1, and hnRNP A2. These four subtypes are generally believed to have similar structures and functions [86]. Among them, the hnRNP A1 protein has been extensively studied, and it plays a crucial role in mRNA metabolism and biosynthesis. It facilitates the regulation of the transcriptional functions of target genes and the alternative splicing of over 25 identified genes [87]. During muscle development, hnRNP A1 primarily regulates the alternative splicing process of multiple genes related to muscle development, thereby influencing cell metabolism. LIU T et al. [88] discovered that hnRNP A1 modulates various muscle development-related genes, such as myocyte enhancer factor 2C (*Mef2c*), leucine rich repeat in FLII interacting protein 1(*LRRFIP1*), ubiquitin specific peptidase 28(*USP28*), and ATP-binding cassette sub-family C member 9 (*ABCC9*). Additionally, studies have indicated that a decreased expression of hnRNP A1 in skeletal muscle inhibits glycogen synthesis, thereby affecting skeletal muscle energy metabolism and systemic insulin sensitivity [89,90]. Furthermore, elevated levels of hnRNP A1 have been associated with a shift in the splicing pattern of type I myotonic dystrophy-related genes towards a fetal splicing pattern, resulting in severe muscle weakness and myopathy [91].

hnRNP A2/B1 is functionally similar to hnRNP A1, but its impact on muscle development differs. It affects muscle development by participating in the transcriptional regulation of relevant genes. Research indicates that hnRNP A2/B1 directly binds to the promoters of *ACTA2* and *TAGLN* genes, thereby regulating the expression of genes associated with smooth muscle development at the transcriptional level [92]. Additionally, studies have shown that hnRNP A2/B1 can regulate stem cell differentiation [93]. WHEELER J and colleagues [94] discovered that hnRNP A2/B1 controls muscle differentiation via the alternative splicing of pre-mRNA of *MBNL1*, *MBNL2*, and *RBFox2* in regenerating myogenic cells, which aligns with the findings of Wang H et al. [93]. Interestingly, hnRNP A2/B1 binding to its own 3′UTR region was also noted in this study, suggesting its potential role in RNA localization and protein translation.

### 3.2. Role of hnRNP M during Muscle Development

hnRNP M contains three RRM motifs which typically function in splicing processes within cells. Previous studies have indicated that the interaction between hnRNP M and mTORC2 can impact the phosphorylation of S422 serine in SGK1, thereby regulating muscle differentiation. The mechanism of differentiation mediated by hnRNP M operates through the SGK1/FoxOs pathway. The authors hypothesized that hnRNP M acts as a mediator, transmitting signals from mTORC2 to SGK1 and FoxO and thereby regulating muscle differentiation [95].

### 3.3. Role of hnRNP E1 during Muscle Development

hnRNP E1, also known as PCBP1, is a versatile regulator of gene expression and protein function. It performs various functions; for example, it acts as a transcription factor, a translation factor, a post-transcriptional regulator, and a molecular chaperone. In a study by Espinoza-Lewis RA et al. [96], the disruption of hnRNP E1 in mouse skeletal muscle satellite cells led to disordered cell proliferation and differentiation. Mutations in hnRNP E1 in mice had significant impacts on muscle growth, slow to fast muscle fiber conversion, and the proliferation of muscle satellite cells. The mechanism behind this primarily involves hnRNP E1 regulating the production of muscle-specific miRNAs (miR-1, miR-133, and miR-206) by forming a complex with AGO 2 and other miRNAs. These miRNAs can then be packaged into extracellular vesicles to exert their function [97]. Studies have shown that miR-1, miR-133, and miR-206 play crucial roles in myocyte proliferation and differentiation [98,99].

### 3.4. Role of hnRNP G during Muscle Development

hnRNP G is a ubiquitous type of hnRNP that is expressed at low levels and closely involved in the processing of mRNA precursors [100]. One study revealed that hnRNP G inhibits the skeletal muscle-specific exon SK of *TPM3* through a specific sequence while stimulating the expression of the non-muscle cell exon NM of *TPM3*. The presence of hnRNP G in cells helps to determine the splicing preference, and its high levels in skeletal muscle prevent the expression of incorrect exons, ultimately preventing skeletal muscle dystrophy [101].

### 3.5. Role of hnRNP L during Muscle Development

In one study, hnRNP L knockout resulted in dose-dependent myotube apoptosis mediated by caspase-3 (the major terminal cleaving enzyme during apoptosis). Zebrafish subjected to hnRNP L knockdown using antisense morpholino oligonucleotide (Morpholino, MO) exhibited abnormal body shapes, bent tails, poor muscle birefringence, and pericardial edema [42]. Similarly, in another study, the downregulation of *smooth*, a homologue of hnRNP L in Drosophila cells, led to severe abnormal muscle formation, impaired movement, and premature death [102]. Moreover, ZHAO Y et al. [103] demonstrated that the myogenic transcription factor MyoD1 produces enhancer RNA (eRNA), which interacts with hnRNP L to regulate the expression levels of adjacent myoglobin and apolipoprotein L6, thereby influencing myogenic differentiation. Furthermore, studies have revealed that hnRNP L can interact with lncRNA THRIL to modulate the level of TNFα, a cytokine that impacts the myogenic differentiation of muscle satellite cells [104].

### 3.6. Role of hnRNP D during Muscle Development

hnRNP D, also known as AUF1 (AU-rich binding factor 1), has been extensively studied for its role as an attenuation factor of ARE-mRNA [105,106]. In a study by Chenette DM et al. [107], it was discovered that hnRNP D binds to the ARE sequence of pre-mRNA and collaborates with other RNA-binding proteins to facilitate the rapid degradation of these pre-mRNA molecules. In muscle tissue, hnRNP D plays a crucial role in targeting and degrading ARE-mRNA to regulate satellite cell function. Another study by Abbadi D et al. [108] revealed that hnRNP D regulates myogenic differentiation by controlling the degradation of ARE-mRNA in muscle satellite cells. Myoblasts contain various types of ARE-mRNA, which function in promoting myoblast development, inhibiting myoblast proliferation, inducing differentiation, and activating mature myotube differentiation mediated through the Sonic Hedgehog signaling pathway. Therefore, hnRNP D plays a pivotal role in coordinating muscle stem cell proliferation, self-renewal, and myoblast differentiation. Additionally, hnRNP D is associated with the 3′UTR of *Mef2c* mRNA and promotes the translation of *Mef2c* [109].

### 3.7. Role of hnRNP H during Muscle Development

The main function of hnRNP H in cells is to enhance or inhibit the splicing of precursor RNAs. Previous studies have reported that hnRNP H plays a role in regulating the splicing of various precursor RNAs [110]. In the rat β-tropomyosin gene, there are two exons (exon 6 and exon 7) that undergo mutually exclusive splicing. Exon 6 is transcribed into mRNA-encoding *TM-1* in non-muscle cells, while exon 7 is transcribed into mRNA-encoding *β-TM* in skeletal muscle cells. The six nucleotides at the 5′ end of β-myosin exon 7 are known as exonic splicing silencers (ESSs). HnRNP H specifically binds to the ESSs, resulting in the excision of exon 7 and promoting cell development towards the non-muscle cell direction [111,112] (Table 1) (Figure 6).

## 4. hnRNPs and Muscle-Related Diseases

hnRNPs play a crucial role in muscle diseases, particularly in the progression of muscle atrophy and myositis. Following skeletal muscle injury, muscle satellite cells are activated and subsequently undergo proliferation and differentiation to form muscle fibers, thereby repairing the damaged muscles. According to modern medicine, muscle weakness and atrophy can be attributed to two main factors. Firstly, strenuous exercise or severe accidents can cause damage to muscle satellite cells and myoblasts, leading to impaired muscle regeneration. Secondly, the proliferative capacity of muscle satellite cells decreases with age. By regulating mRNA stability, hnRNPs can significantly contribute to skeletal muscle growth and muscle injury repair. In one study, mice lacking the hnRNP D gene experienced accelerated skeletal muscle atrophy with age, known as sarcopenia [107,122]. Mice have also been shown to exhibit severe impairment in skeletal muscle regeneration after injury and develop limb-girdle muscular dystrophy (LGMD), which affects the extremities and upper chest. In humans, mutations in the hnRNPDL gene are associated with LGMD type 1G disease [123,124]. Myotonic dystrophy type I (DM1) is caused by the isolation of the RNA-binding protein MBNL1 by the CUG repeat spacer, resulting in the missplicing of downstream targets. It has been discovered [42] that the overexpression of hnRNP L can alleviate the toxicity caused by the RNA produced by the CUG repeat spacer.

Sporadic inclusion body myositis (sIBM) is a degenerative muscle disease that is not well understood. In a recent study, researchers observed changes in the subcellular localization of hnRNP A1 and hnRNP A2/B1 in sIBM patients compared to normal individuals. Specifically, there was a decrease in the expression of hnRNP A1 and hnRNP A2/B1 in the myonuclei of sIBM patients, while their expression increased in the sarcoplasm of the muscle. Additionally, the formation of prion-like structures [125] was observed in borderline vacuolar myopathies associated with mutations in hnRNP A1 and hnRNPA2/B1, such as oculopharyngeal muscular dystrophy [126,127]. This study suggests that sIBM muscle fiber injury involves the formation and aggregation of pathological ribonucleoprotein granules through prion-like structures, leading to toxicity and the disruption of RNA homeostasis [125,126].

Multisystem proteinopathy (MSP) is a rare genetic disease that affects multiple systems in the body, including the bone, muscle, and nervous systems. Recent studies have shown that the disease is associated with specific prion-like domains known as PrLD in proteins hnRNP A2/B1 and hnRNP A1. Pathogenic mutations in these proteins result in the strengthening of the ‘steric zipper’ motif of PrLD [128], causing an accumulation of hnRNP A2/B1 and hnRNP A1 in the affected areas. Additionally, other proteins, such as TDP-43 and FUS, which also contain PrLD, have been found to accumulate at the lesion site [126].

Spinal muscular atrophy (SMA) is an autosomal recessive myopathy caused by mutations in the *SMN1*, resulting in reduced SMN protein levels. In the splicing process of *SMN2* pre-mRNA, hnRNP A1 competitively binds to the 5′ splice site, preventing recognition by U1 snRNP. As a result, the splicing complex fails to recognize exon 7, and it is removed from the pre-mRNA transcript [129]. Researchers have designed Antisense oligonucleotide drugs targeting the hnRNP A1-dependent splicing silencer ISS-N1 located in intron 7 of *SMN2* pre-mRNA. These drugs aim to replace the hnRNP protein at the ISS-N1 site on *SMN2* pre-mRNA, thereby inhibiting its binding to *SMN2* pre-mRNA. This approach is being explored for the treatment of SMA [130,131]. Studies have also indicated that hnRNP G and hnRNP R can bind to the AU-rich element at the 3′ end of the SMN exon, inhibiting splicing and leading to the removal of exon 7 [132]. Additionally, hnRNP R plays a role in SMN transport within cells [133] and may be associated with SMA disease.

## 5. Conclusions and Outlook

Heterogeneous nuclear ribonucleoproteins (hnRNPs) play a significant role in muscle development through various pathways. Currently, most research on the involvement of hnRNPs in muscle development focuses on analyzing their molecular functions. However, due to the complexity of the muscle development process and the challenges in conducting research, the molecular mechanism of hnRNPs in muscle development remains unclear. The recent advancements in experimental methods, such as clip-seq, have the potential to shed more light on protein and RNA interactions and transcriptional regulation. Therefore, it is important to further investigate the mechanism of action of hnRNPs in the occurrence of myopathy.

## Figures and Tables

**Figure 1 biomolecules-13-01434-f001:**
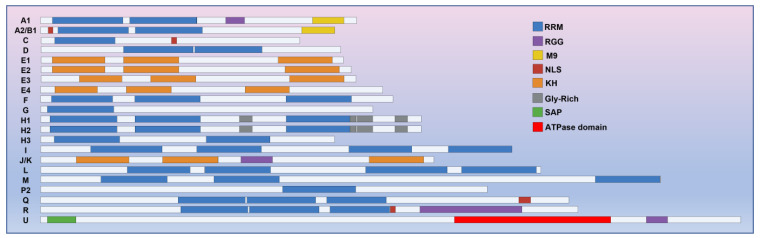
hnRNPs consist of three types of RNA binding motifs: RRM, RGG, and KH domains. The RRM motif is the most prevalent among these motifs in hnRNPs. Apart from RNA-binding motifs, certain hnRNPs may also possess auxiliary domains, namely the M9, NLS, SAP, and ATPase domains. These auxiliary domains play crucial roles in various processes such as nucleocytoplasmic shuttling, DNA binding, and the oligomerization of hnRNPs with caRNAs.

**Figure 2 biomolecules-13-01434-f002:**
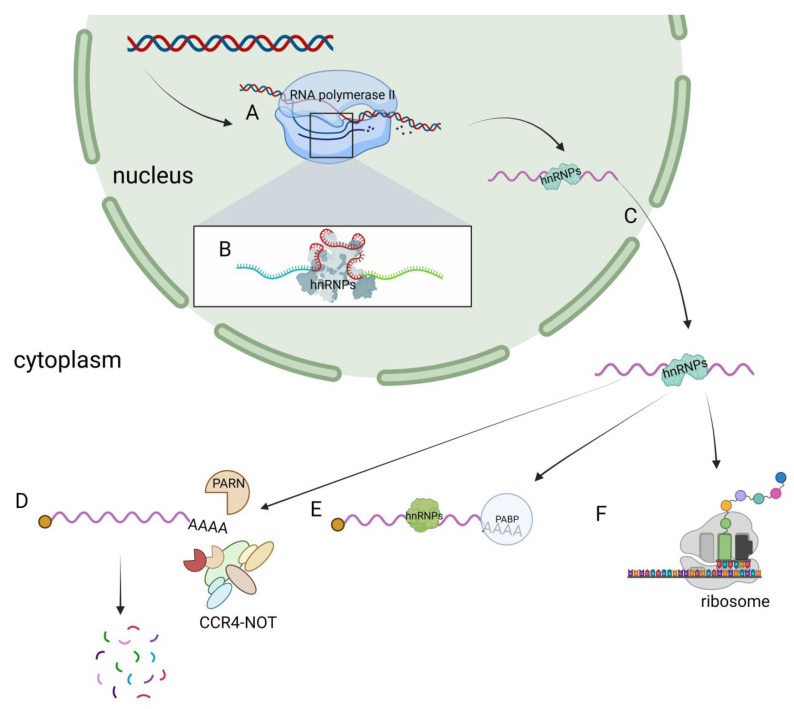
Functions of hnRNPs in cells. (**A**) hnRNPs have transcriptional regulatory functions. (**B**) hnRNPs regulate the alternative splicing of pre-mRNA. (**C**) The nucleocytoplasmic shuttling function of hnRNPs. (**D**) hnRNPs are involved in the degradation of mRNA. (**E**) hnRNPs participate in the regulation of mRNA stability. (**F**) hnRNPs regulate the translation process.

**Figure 3 biomolecules-13-01434-f003:**
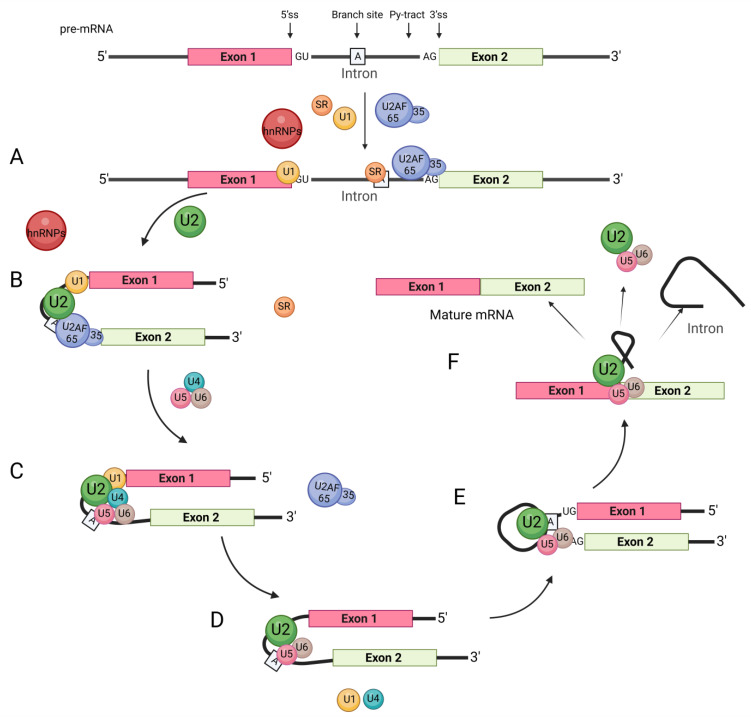
General splicing process. The regulation of RNA-binding proteins such as hnRNPs and SRs involves several steps. (**A**) The splicing factor U1 snRNP complex binds to the 5′ splice site, U2AF65 binds to the polypyrimidine site (Py-tract), and U2AF35 binds to the AG bases at the 3′ splice site. The U2AF heterodimer combines with the 3′ splicing site to recognize the intron splicing signal. (**B**) U2 snRNP binds to the branch site with the assistance of U2AF. (**C**) snRNP U4, U6, U5 join the complex, while U2AF dissociates from the complex. (**D**) U1 snRNP and U4 snRNP subsequently leave the complex through a series of conformational transitions. (**E**) The first transesterification reaction connects the 5′ss to the branch site and cleaves the RNA strand, forming a lariat structure. (**F**) In the second transesterification reaction, the exons are ligated to each other to form the mRNA, and the introns are released in a lariat structure.

**Figure 4 biomolecules-13-01434-f004:**
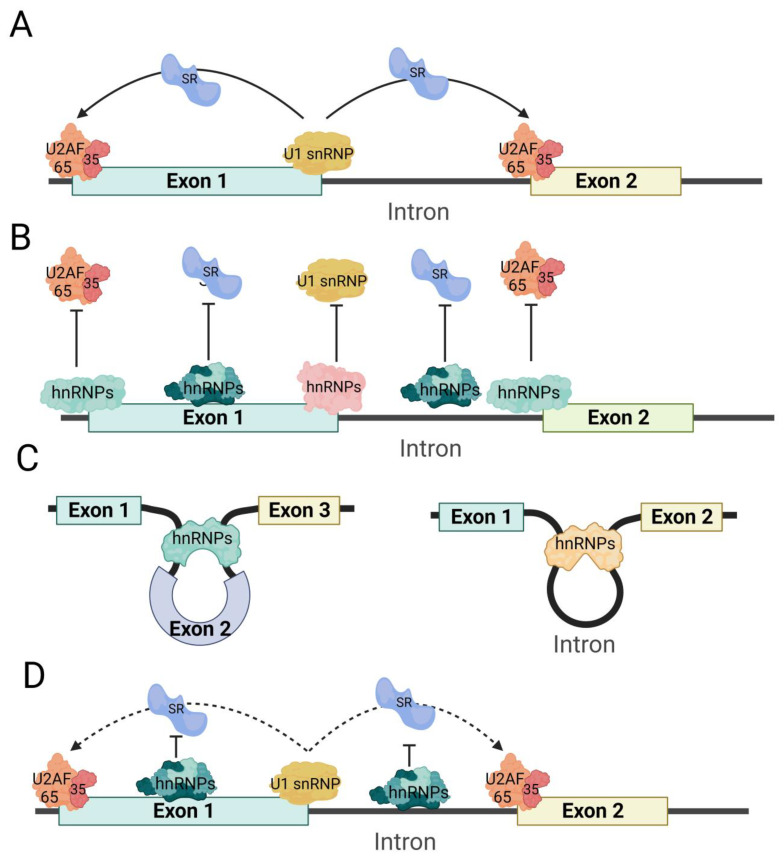
Schematic diagram of hnRNPs regulating variable splicing. The process of exon excision (left) and intron excision (right) shown in panel (**A**) relates to when hnRNPs are not involved in variable splicing. Panel (**B**) demonstrates how hnRNPs prevent splicing factors such as U2AF heterodimer and U1 snRNPs from binding to pre-mRNA. This prevention is achieved through the recognition of splicing sites and masking effects. In panel (**C**), hnRNPs are shown to promote exon (left) and intron (right) retention by binding to pre-mRNA, forming a special pre-mRNA loop structure. This structure prevents splicing factors from recognizing splice sites on pre-mRNA. Finally, in panel (**D**), hnRNPs bind to SRs and hinder the promotion of splicing by SRs.

**Figure 5 biomolecules-13-01434-f005:**
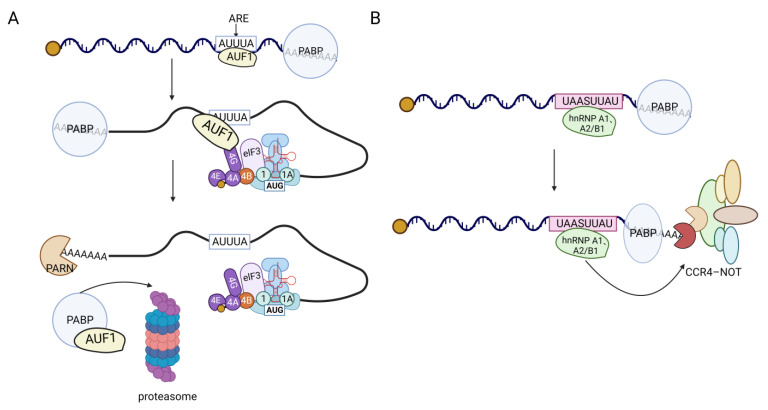
hnRNPs-mediated mRNA degradation mechanism. (**A**) In the translation process, eIF4G binds to AUF1, promoting the dissociation of AUF1 from the ARE sequence. The released AUF1 then binds to PABP, leading to the exposure of the polyadenine nucleic acid tail for mRNA degradation. Both AUF1 and PABP are subsequently degraded through the proteasome. (**B**) hnRNP A2/B1 and hnRNP A1 interact with the UAASUUAU sequence located in the 3′UTR of mRNA. This interaction plays a role in controlling mRNA deadenylation, which is carried out by the CCR4-NOT deadenylation complex. Ultimately, this process influences the degradation of the transcript.

**Figure 6 biomolecules-13-01434-f006:**
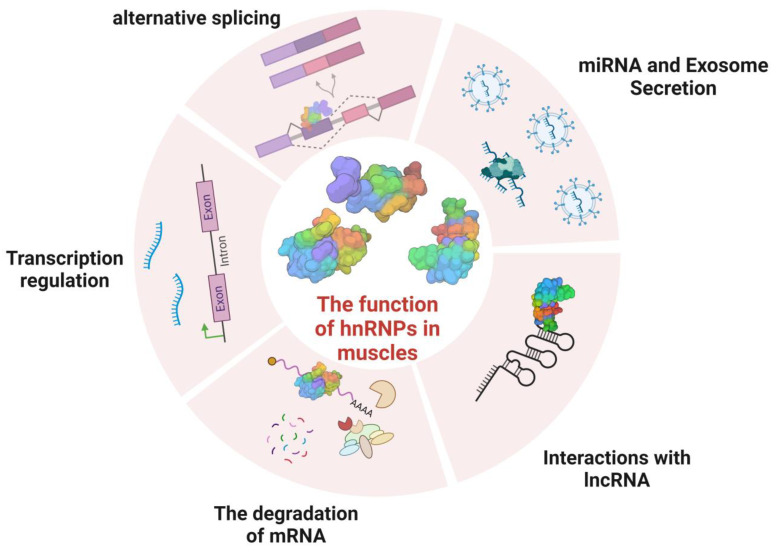
The function of hnRNPs in muscles. This figure illustrates the role of heterogeneous nuclear ribonucleoproteins (hnRNPs) in the regulation of muscle growth and development. hnRNPs exert their influence through various mechanisms, including the regulation of alternative splicing, miRNA synthesis, interaction with long non-coding RNA (lncRNA), the regulation of mRNA degradation, and transcription regulation.

**Table 1 biomolecules-13-01434-t001:** Relationship between hnRNPs and muscle development.

Protein	Domain/Functional Motif	Preferred Binding Sequence	Reported Function	Report Function in Muscle	References
A1	2×RRM Gly rich RCG	UAGGG(A/U)	Alternative splicing, telomere biogenesis, mRNA stability, translation regulation	Key genes undergo alternative splicing, affecting energy metabolism	[87,88,89,113,114]
A2/B1	2×RRM Gly rich RCG	(UUAGGG)N	Alternative splicing, mRNA stability regulation, RNA localization	Transcriptional Regulation of relevant genes	[91,113,115]
M	3×RRM	G or U rich	Alternative splicing	Protein modification	[9,95,116]
E1 (PCBP1)	3×KH	C rich	Transcriptional translation multi-level regulation, mRNA stability, alternative splicing	Regulating exocrine secretion, regulating muscle miRNA production	[96,101,117]
G	RRM Gly rich	CC(A/C)	Alternative splicing, mRNA precursor processing	Determining variable splicing preferences	[100,101]
L	4×RRM Gly rich	CA repeat	Alternative splicing, mRNA stability	Adjustment of myogenic differentiation, co-regulation of key cytokine levels with lncRNA	[102,103,118,119]
D (AUF1)	2×RRM	AU rich	RNA decay, telomere maintenance	ARE-mRNA decay	[105,106,107,108,109,120]
H	3×RRM	GGGA	Alternative splicing, polyadenylation	Alternative Splicing of key genes	[110,111,121]

## Data Availability

Not applicable.

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
