# Peer review of "Research Progress on the Structural and Functional Roles of hnRNPs in Muscle Development"

_biomolecules, 2023, doi:10.3390/biom13101434_

Round 1
Reviewer 1 Report
The paper delves into the role of heterogeneous nuclear ribonucleoproteins (hnRNPs) in muscle development, shedding light on their structural and functional significance. It highlights the multifaceted involvement of hnRNPs in regulating various aspects of RNA metabolism, including alternative splicing, transcription, miRNA regulation, and mRNA stability. Additionally, it explores the potential link between hnRNPs and muscle-related diseases. While this paper offers valuable insights into the field of hnRNPs in muscle biology, there are several areas where clarification and refinement are necessary for a more comprehensive understanding.
Comments and Suggestions for Improvement:
Figures: In Figure 1, the size of the figures is too small for comfortable reading. Please provide higher-resolution and larger images. Also, ensure that you explain all the abbreviations of RNA domain names on the right. Some colors used in the figure are too similar; consider using distinct subset of colors for RNA-binding motifs and auxiliary domains.
Figure 2: Label the complex names in each panel for clarity. Additionally, please label nucleus and cytoplasm.
Figure 3: Provide a detailed explanation of this process in the figure legend to aid in comprehension.
Overall, this paper holds promise for advancing our understanding of hnRNPs' roles in muscle development and their potential relevance to muscle-related diseases. Addressing the above-mentioned points will significantly enhance the clarity, rigor, and impact of the research.
Minor editing of English language required
Reviewer 2 Report
The authors present a review of the structure and function of a family of proteins - the HNRNPs - in relation to muscle development. These proteins are of growing interest as more and more disease causing mutations are identified.
As it is a review the main challenge for the authors is to adequately cover all the relevant data without drowning the reader in tedious, minor details. In all, I think the authors have done a decent job in summarizing the current status. But, as always, I have some comments that I hope the authors will consider:
Major comments:
a) Lines 137-8 references ref50, but ref50 show that knock-down of HNRNPA1 cause increased inclusion of that exon (legend of Figure 1 in ref 50 states "Reduced expression of hnRNP A1 results in increased alternative exon 10 inclusion of Tau pre-mRNA"). So opposite of the current statement in the manuscript that hnRNP A1 promotes inclusion. I sincerely hope there are no other such cases.
b) the discussion of reported pathogenic variants are quite light (Chapter 4). There are many other reports of mutations in several of these genes. So I'd like this section expanded and updated.
c) I think the phraseology on line 396 is strange. Nusinersen is not directly targeted to hnRNPA1, but towards the SMN2 gene. It is binding to the SMN2 pre-mRNA, which then prevents the splice-blocking effect of hnRNAs (as their binding site is already occupied by the oligonucleotide) leading to increased inclusion of exon 7 of SMN2. The role of hnRNPs in other genes is (in theory at least) not affected, which it clearly would if hnRNP was the direct target.
Minor comments:
1) The reference list is not in the recommended style and contains some errors. In ref 52 the initials of the first author is registered as individual authors. Ref 126 and 127 are the same (with page numbers missing in one), and so are refs 132 & 133. Please take a careful look at the references.
2) Gene names should be in italics.
3) In chapter 2.2. it is not stated directly that hnRNPs primarily bind single stranded RNA and in some special cases single stranded DNA. I think a reader trying to get a grip on this protein family would like to be told this specifically.
4) Line 49: to me "preliminary insights into the sequence" is strange. I don't think those would change or expand much. I do agree that there are more discoveries to be made regarding function of this important family.
5) Line 70: This should be rewritten. Now it says that "RGG is responsible for the RNA-binding activity of hnRNPs" contradicting the RRM-domain paragraph. I don't think that is what the authors want to tell.
6) Line 79: I think this should be in plural "These interactions" as there is more than one bond between these residues and the RNA.
7) Line 97-8: This sentence is somewhat loose and it is already mentioned as well. It is not unique to hnRNPA1 either as all the other hnRNPs with two RRMs use both of these to bind RNA.
8) Figure 1: Ref 13 states that hnRNP U has an RGG-domain. It is not mentioned in the figure
9) All the figures are visually nice, but the legend of Figure 2 should be expanded (or more text in the figure) to explain the various components.
10) Line 127: Why "fragments near pre-mRNA splice sites"? I think it binds "sequence motifs" or similar - not fragments.
11) Tau (around lines 137) is a protein. The gene is (currently) called MAPT. I think it is worthwhile to mention the gene names especially when they are very different from the protein, or old gene, name. This is similar for Smαa and Sm22α that also have modern gene names (ACTA2 and TAGLN).
12) Line 144 mentions RBD3 and RBD4 domains. Should be RRM3 & 4.
13) Line 155: "C175G pre-mRNA" is not a gene and should be briefly explained for readers that have not already studied the reference.
14) Figure 3: This fig illustrates splicing and not alternative splicing as now stated.
15) Line 383: MSP is most often described as "Multisystem proteinopathy". I see no reason to not sticking to that term
16) Line 392: SMA is caused by mutations in SMN1.
Very minor, but still worth fixing: there are several missing spaces after full stop (.) and extra commas (,) and other small grammatical errors. Like, line 384 "skeletal, muscle, and nervous systems" should be "skeletal muscle and nervous systems". Line 139 does not have a space before "Similarly". And "the AG base" on line 124 should be bases (in plural) as they are two (or base pair if preferred), "hnRNPs proteins" on line 127 should be "hnRNP proteins" etc.
In conclusion I look forward to seeing the corrected manuscript published.
The English is in general good, but there are some spelling mistakes and more worryingly some misunderstandings from the literature. This is covered in the general scientific comment section above.
